# Accelerated single cell seeding in relapsed multiple myeloma

Heather J. Landau[1,2,12], Venkata Yellapantula[3,4,12], Benjamin T. Diamond[4], Even H. Rustad [4],
Kylee H. Maclachlan[4], Gunes Gundem[3], Juan Medina-Martinez[3], Juan Arango Ossa[3], Max F. Levine [3],
Yangyu Zhou[3], Rajya Kappagantula[5], Priscilla Baez[5], Marc Attiye [5], Alvin Makohon-Moore[5], Lance Zhang[5],
Eileen M. Boyle[6], Cody Ashby [7], Patrick Blaney[6], Minal Patel[8], Yanming Zhang[9], Ahmet Dogan [10],
David J. Chung[1,2], Sergio Giralt[1,2], Oscar B. Lahoud[1,2], Jonathan U. Peled[1,2], Michael Scordo[1,2], Gunjan Shah[1,2],
Hani Hassoun[2,4], Neha S. Korde[2,4], Alexander M. Lesokhin[2,4], Sydney Lu[2,4], Sham Mailankody[2,4],
Urvi Shah [2,4], Eric Smith[2,4], Malin L. Hultcrantz [2,4], Gary A. Ulaner[11], Frits van Rhee[7], Gareth J. Morgan[6],
Ola Landgren [2,4,13], Elli Papaemmanuil [3,13], Christine Iacobuzio-Donahue [5,13] & Francesco Maura [4,13✉]

Multiple myeloma (MM) progression is characterized by the seeding of cancer cells in different anatomic sites. To characterize this evolutionary process, we interrogated, by whole genome sequencing, 25 samples collected at autopsy from 4 patients with relapsed MM and an additional set of 125 whole exomes collected from 51 patients. Mutational signatures analysis showed how cytotoxic agents introduce hundreds of unique mutations in each surviving cancer cell, detectable by bulk sequencing only in cases of clonal expansion of a single cancer cell bearing the mutational signature. Thus, a unique, single-cell genomic barcode can link chemotherapy exposure to a discrete time window in a patient's life. We leveraged this concept to show that MM systemic seeding is accelerated at relapse and appears to be driven by the survival and subsequent expansion of a single myeloma cell following treatment with high-dose melphalan therapy and autologous stem cell transplant.

[1] Adult Bone Marrow Transplant Service, Department of Medicine, Memorial Sloan Kettering Cancer Center, New York, NY, USA. [2] Department of Medicine, Weill Cornell Medical College, New York, NY, USA. [3] Department of Epidemiology and Biostatistics, Memorial Sloan Kettering Cancer Center, New York, NY, USA. [4] Myeloma Service, Department of Medicine, Memorial Sloan Kettering Cancer Center, New York, NY, USA. [5] Human Oncology & Pathogenesis Program, Memorial Sloan Kettering Cancer Center, New York, NY, USA. [6] NYU Perlmutter Cancer Center, New York, NY, USA. [7] Myeloma Center, University of Arkansas for Medical Sciences, Little Rock, AR, USA. [8] Center for Hematological Malignancies, Department of Medicine, Memorial Sloan Kettering Cancer Center, New York, NY, USA. [9] Cytogenetics Laboratory, Department of Pathology, Memorial Sloan Kettering Cancer Center, New York, NY, USA. [10] Hematopathology Service, Department of Pathology, Memorial Sloan Kettering Cancer Center, New York, NY, USA. [11] Department of Radiology, Memorial Sloan Kettering Cancer Center, New York, NY, USA. [12] These authors contributed equally: Heather J. Landau, Venkata Yellapantula. [13] These authors jointly supervised this work: Ola Landgren, Elli Papaemmanuil, Christine Iacobuzio-Donahue, Francesco Maura. ✉email: mauraf@mskcc.org

The pathogenesis of multiple myeloma (MM) is characterized by a long and complex evolutionary process through two clinically defined precursor stages: monoclonal gammopathy of uncertain significance and smoldering MM[1–4]. The progression of precursor disease to invasive MM is characterized by branching evolutionary patterns and clonal sweeps together with local evolution and expansion of cancer cells in varying anatomical sites[5,6]. Divergence and progression at distinct sites of disease—spatial evolution—magnifies the genomic heterogeneity of MM, where a range of clones compete for dominance and are positively selected according to their genetic driver landscape, reflected in their ability to best adapt to the local environment[7–12]. Similar evolutionary patterns have also been observed in patients with MM at clinical relapse[13–15]. However, it is unclear if, at the time of relapse, the new disease sites reflect a pre-existing but previously undetected disease localization or a new dissemination of disease seeding. Furthermore, despite the unquestionable spatial heterogeneity of MM[13], to our knowledge, its development over time has not been investigated in a systematic manner.

Somatic mutations in cancer genomes are caused by different mutational processes, each of which generates a characteristic mutational signature[16,17]. Considering each single nucleotide variant (SNV) together with its neighboring bases at 5′ and 3′ (the trinucleotide context), more than 40 mutational signatures [or single base substitution (SBS) signatures] have been described, some of which are associated with defective DNA repair

mechanisms, exposure to exogenous carcinogens, or radical oxygen stress[16,17]. Using large genomic datasets, we recently described the landscape of mutational processes active in MM[8,18–22]. At diagnosis (i.e., prior to therapy), the mutational landscape is shaped by seven main mutational processes, five of which have a recognized etiology: activation-induced cytidine deaminase (AID; SBS9), aging (SBS1 and SBS5), and apolipoprotein B mRNA Editing Catalytic Polypeptide-like (APOBEC; SBS2 and SBS13). At relapse, we reported a new mutational signature associated with melphalan exposure named SBS-MM1[21]. Similar to other chemotherapy-related mutational signatures described in other malignancies, each surviving myeloma cell exposed to melphalan will acquire a unique set of mutations detectable by bulk sequencing only if a melphalan-exposed single cell is positively selected and expands (Fig. 1a)[17,23,24]. As a consequence, two different MM localizations after melphalan exposure can present three different scenarios. In the first, all cancer cells in both localizations will share an identical catalogue of melphalan-related mutations, suggesting that both anatomic sites were seeded by one cancer cell surviving the exposure to melphalan (Fig. 1b). In the second each localization will have a unique catalogue of melphalan-related mutations, suggesting that cells at the two localizations pre-existed the exposure to melphalan (Fig. 1c). Finally, it is possible that myeloma cells are reintroduced with the stem cell infusion during autologous transplant, avoiding in this way any exposure to melphalan and the associated mutational signature[21]. Although the impact of

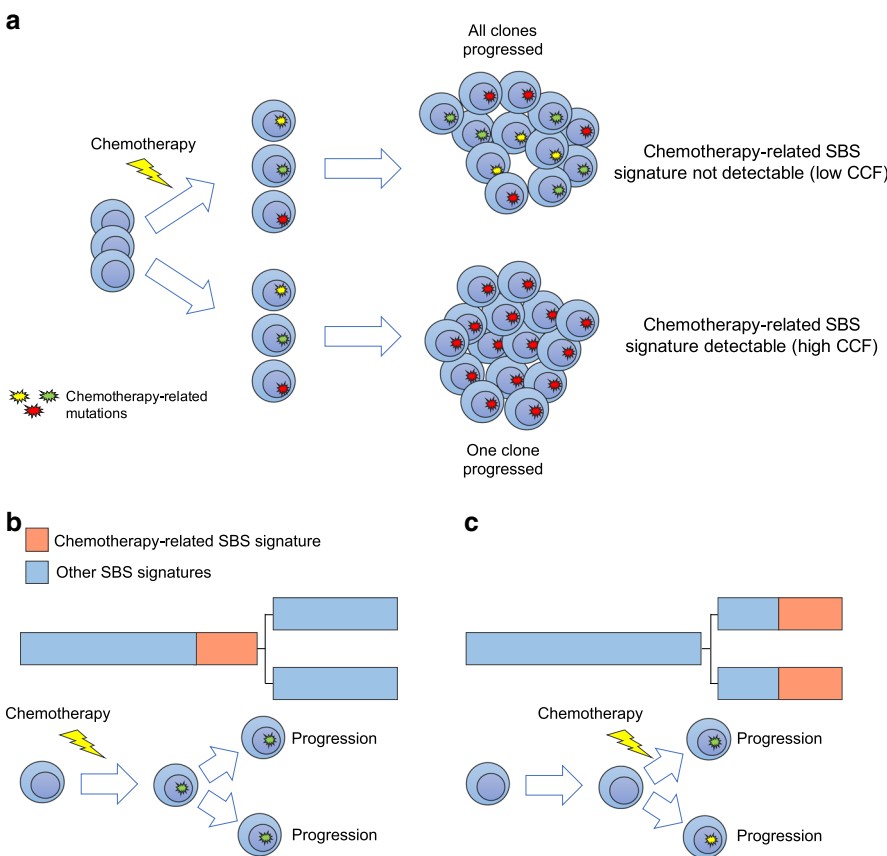

**Fig. 1 The development of chemotherapy-related mutational signatures. a** A schema summarizing the single cell expansion model. In this model, chemotherapy-related mutational signatures will be detectable only if one cancer cell is selected and takes the clonal dominance. (SBS = single base substitution; CCF = cancer-cell fraction). **b**, **c** Two possible scenarios for the development of chemotherapy-related mutational signatures in two different disease localizations. In (**b**), chemotherapy is delivered to the trunk of the phylogenetic tree prior to any branching, while in (**c**) chemotherapy is delivered after branching. The phylogenetic tree trunk and branch lengths represent the mutational load.

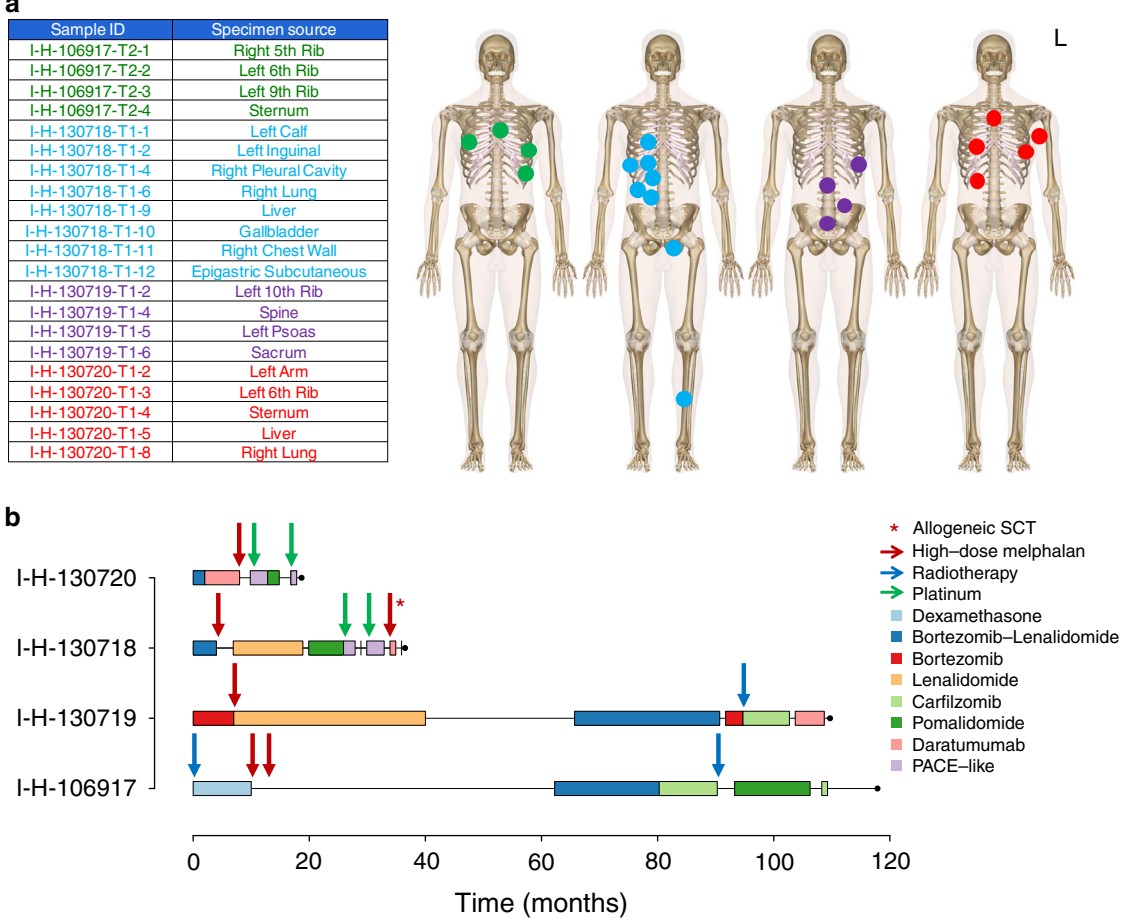

**Fig. 2 Patient cohort and samples. a** Anatomical sites that were biopsied in each patient. L = left. **b** Summary of the treatment history of each patient. After the front-line therapy, only agents new to each patient were reported. Red, blue, and green vertical arrows represent the time at which patients were exposed to high-dose melphalan and autologous transplant, radiotherapy and platinum-based chemotherapy, respectively. For I-H-130718, the second high-dose melphalan exposure was an allogeneic stem cell transplant which is annotated with a red asterisk. (PACE = cisplatin, doxorubicin, cyclophosphamide and etoposide; SCT = stem cell transplant).

these chemotherapy-related mutations on cancer aggressiveness at clinical relapse is not fully understood, chemotherapy-related mutational signatures represent a unique single-cell genomic barcode for clonal cells derived from a single propagating cell linked to a discrete time point in each patient's life.

Here, to investigate the spatial and temporal systemic dissemination of MM at clinical relapse, we interrogated 25 samples collected at warm autopsy from four patients with relapsed/refractory MM by whole-genome sequencing (WGS). Leveraging the chemotherapy-related mutational signatures caused by melphalan and platinum-based chemotherapy, we show that disease seeding is accelerated at clinical relapse and appears to be driven by a single myeloma propagating cell that survives after high-dose melphalan therapy followed by autologous stem cell transplant.

## Results

**Phylogenetic trees and disease seeding**. To investigate the spatial and temporal systemic dissemination of MM, we investigated the WGS profile of twenty-one tumor and four non-tumor samples collected from four patients (Fig. 2; Supplementary Tables 1–2). Normal samples were taken from uninvolved skeleton muscle. All patients consented to autopsy and sample collection as a part of the Last Wish Program at Memorial Sloan

Kettering Cancer Center (MSKCC)[25]. An additional cohort of 125 published whole exomes collected from different disease localization in 51 patients was included (Supplementary Tables 3–4)[13]. To define the key evolutionary trajectories and drivers involved in MM systemic seeding, we reconstructed the clonal and subclonal composition of each patient included in the WGS and whole-exome sequencing (WXS) cohorts using the Dirichlet process (DP) (see Genomic analysis and Validation set paragraphs, the "Methods" section). Mutations shared as clonal by all samples from the same individual composed the trunk of the phylogenetic tree. Different late clonal or subclonal clusters could have arisen either directly from the trunk of the phylogenetic tree or from one of its branches. To define the evolutionary history of each patient's tumor, we reconstructed the most likely phylogenetic tree solution for each patient and defined the main evolutionary trajectories drawing a line from the tip of each branch, via any larger branches and down through the trunk, following the pigeonhole principle (Fig. 3 and Supplementary Figs. 1–2)[21,26]. A median of 10,938 (range 6977–13,239) mutations were detected by WGS. The trunk of the two longer-surviving myeloma patients in the WGS cohort accounted for 80.5 and 69% of all mutations. In contrast, the trunk was shorter and comprised of far fewer mutations for the two patients with shorter survival (34 and 22% of total

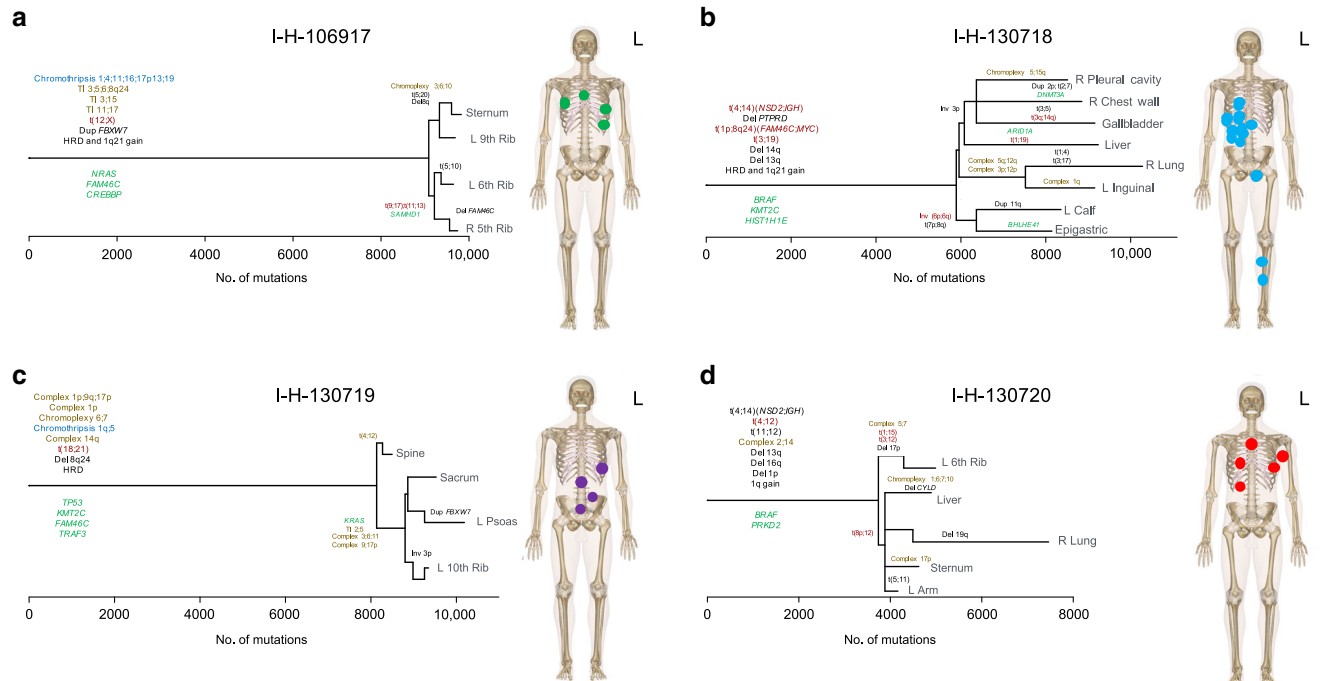

**Fig. 3 Tumor phylogenies.** Phylogenetic trees generated from the Dirichlet process analysis were drawn such that the trunk and branch lengths were proportional to the (sub)clone mutational load, **a** I-H-106917, **b** I-H-130718, **c** I-H-130719 and **d** I-H- 130720. All main drivers (CNA, copy number aberrations; SNV, single nucleotide variants and SV, structural variants) were annotated according to their chronological occurrence and colored according to the type of event. Known driver SNVs were annotated in green, single SVs and CNAs in black (HRD = hyperdiploid), translocations associated with copy number changes in dark red, chromothripsis in blue, other complex events in dark yellow, (TI = templated insertion). Lines from different subclone branches are separated by hooks. Patients with short survival are positioned on the right (I-H-130718 and I-H-130720). L = left.

mutational burden). This difference may be partially explained by the number of samples from each patient, but also by the major subclonal diversification observed in the patients with a short survival. Interestingly, each disease site in the WGS cohort showed a unique set of genomic drivers and aberrations, reflecting distinct evolutionary trajectories and (sub)clonal selection and expansion (Figs. 3 and 4; Supplementary Data 1). The majority of these aberrations were single and complex structural variants (SVs) and copy number aberrations (CNAs), confirming their critical importance in MM progression and evolution[14,27,28]. In line with previous observations, chromothripsis and templated insertion events were often observed in the trunk, while chromoplexy tended to occur in the branches (Figs. 3 and 4)[14]. All patients had at least one clone with a mutation involving the RAS pathway. In contrast to newly diagnosed MM, mutations in known driver genes[20] were rarely identified in the branches of the phylogenetic tree.

Comparing the phylogenetic tree structure and subclonal diversification of the WGS and WXS cohorts, we observed a higher number of evolutionary trajectories in the former. This could be explained by the larger number of specimens from each patient in the WGS series (Supplementary Fig. 3a), reflecting the high spatial heterogeneity of MM detectable in all patients included in this study. Across both series we observed a median of 57-nonsynonymous SNVs per patient (range 21–464). Interestingly, patients with relapsed disease showed a higher number of nonsynonymous SNVs compared with baseline (both WGS and WXS) (Supplementary Fig. 3b). This higher frequency was independent of the number of subclones and the coverage (which was corrected for sample ploidy and purity) (Supplementary Fig. 3c).

**Mutational signatures landscape**. The mutational signature profile of each sample was defined using our recently published workflow[18]. First, we performed de novo extraction of mutational signatures running *SigProfiler* (Supplementary Fig. 4; Supplementary Table 5)[17]. In addition to the eight known MM mutational signatures[18,20,21], we extracted a new signature similar to the recently reported SBS35. This signature has been shown to be associated with exposure to platinum-based chemotherapy[17,23,24,29], a drug class often included in intensive MM chemotherapy regimens. To confirm the presence of each extracted mutational signature and to estimate their contribution to the overall mutational profile, we ran our recently developed fitting algorithm (*mmsig*; Fig. 4). In the WGS cohort, SBS-MM1 and its characteristic transcriptional strand bias were observed all patients, consistent with our prior report[27]. As expected SBS35 was identified only in the two patients who received a platinum-based treatment (I-H-130718 and I-H-130720). In the WXS cohort, SBS35 and SBS-MM1 were observed only when examining mutations acquired or selected after treatment (Fig. 4 and Supplementary Fig. 5). These data suggest that, similarly to other cancers, the mutational landscape of clinically relapsed MM is heavily shaped by exposure to distinct chemotherapeutic agents, such as melphalan or platinum.

To investigate the impact of these chemotherapy-related mutations on the MM genomic profile, we combined the WGS and WXS cohorts and estimated the contribution of both SBS35 and SBS-MM1 among nonsynonymous SNVs[24]. Interestingly, 25.7% (CI 95% 20–32%) of all nonsynonymous mutations at clinical relapse were caused by one of these two mutational processes, suggesting that chemotherapy exposure might

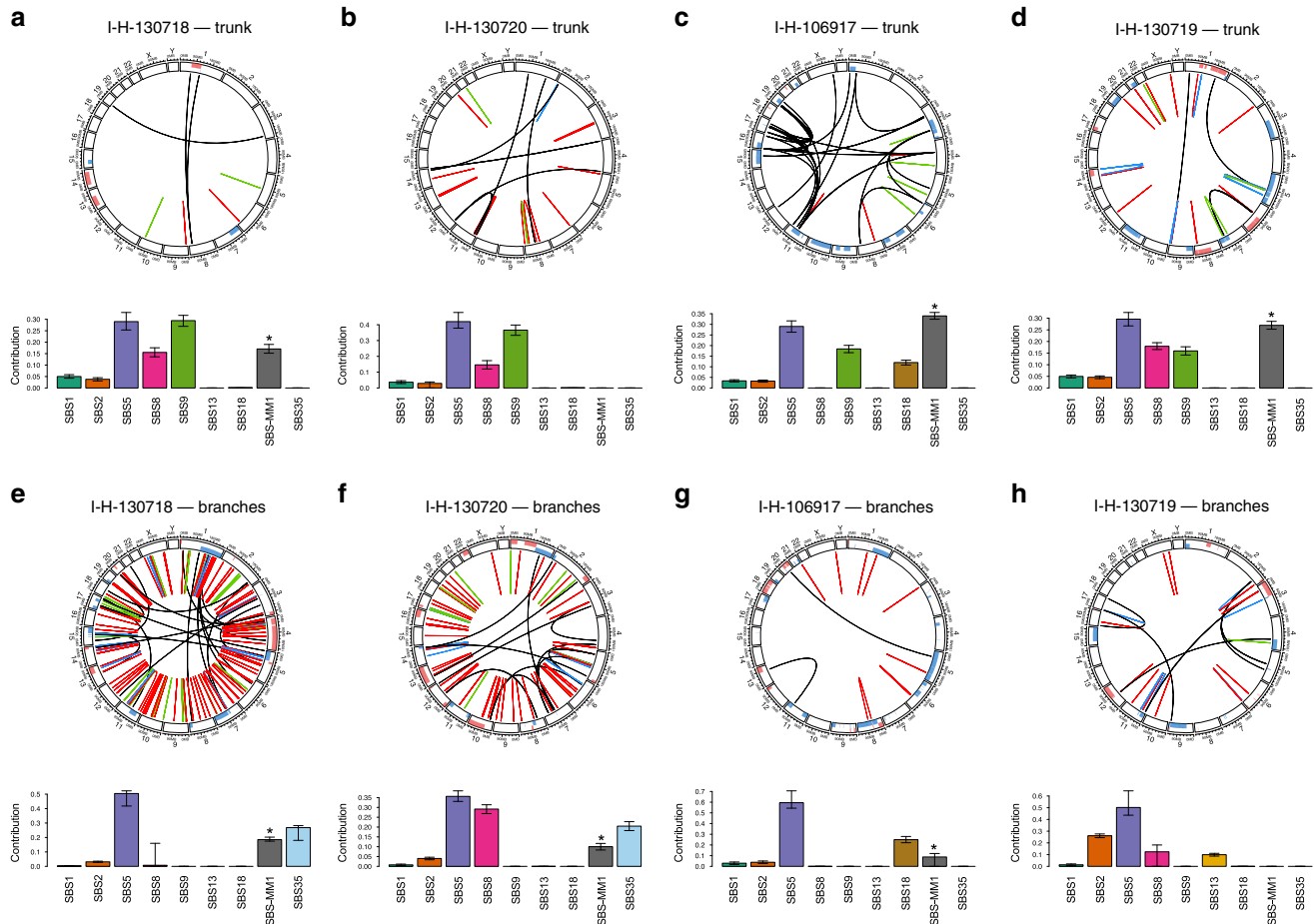

**Fig. 4 Genome plot and mutational signature landscape of all four patients included in this study.** The plots in the top row (**a–d**) show all the genomic events and mutational signatures shared by all samples in each patient (i.e., events in the trunk of the phylogenetic tree). The plots on the bottom row (**e–h**) show the events and mutational signatures not shared by all samples in each patient (i.e., events in the branches of the phylogenetic tree). Copy number aberrations are annotated in the periphery of the circos plots (blue = gain; red = loss of heterozygosity). Structural variants are reported within the circle (black = translocations; blue = inversions; green = tandem-duplications; red = deletions). The asterisks in the barplots reflect the presence of transcriptional strand bias for SBS-MM1 (melphalan-associated signature). Samples I-H-130718 and I-H-130720 were taken from the patients with short survival. Confidence interval of each mutational signature was generated by drawing 1000 mutational profiles from the multinomial distribution, each time repeating the signature fitting procedure, and finally taking the 2.5th and 97.5th percentile for each signature.

play a role in increasing genomic complexity, which has been associated with clinically aggressive disease at relapse (Fig. 5).

To reconstruct the timeline of mutational processes for each patient, we ran *mmsig* on each DP cluster of mutations (Fig. 6a). In line with AID activity early in disease development, SBS9 was detected mostly in the trunk of the phylogenetic trees, while APOBEC activity was detected in both clonal and subclonal clusters[4,8,20,21]. SBS35 was only detected in the branches of the two patients who relapsed post-platinum-based therapy. SBS-MM1 showed a heterogenous landscape. In I-H-106917, SBS-MM1 was detected in the trunk and in all first level branches, but not in the second and third level ones (Fig. 6b). This profile is consistent with a melphalan signature common to all cells, with another subset of melphalan-induced mutations accrued from a second exposure. In I-H-130719, all SBS-MM1-associated mutations were assigned to the trunk, consistent with exposure to high-dose melphalan therapy followed by autologous stem cell transplant received as part of the administered front-line therapy. I-H-130718 was one of the two cases with short survival and platinum exposure. SBS-MM1 was

detected in the trunk and in all the latest branches, reflecting the front-line high-dose melphalan therapy followed by autologous stem cell transplant as well as the allogeneic stem-cell transplant with melphalan-containing conditioning being used at clinical relapse (Fig. 6c). In these three patients, SBS-MM1 was detected in the trunk, consistent with the expansion and the clonal dominance of a single cell surviving the melphalan-exposure. In fact, the large number of SBS-MM1-related mutations shared by all different sites can only be explained by the existence of a recent common ancestor selected after high-dose melphalan therapy followed by autologous stem cell transplant (Fig. 1a–b). These data also show that in these three patients, differences between anatomic sites were not associated with pre-existing undetectable subclones, but by the dissemination from a single MM propagating cell that had survived high-dose melphalan therapy followed by autologous stem cell transplant. This model is also supported by positron emission tomography/computed tomography (PET/CT) imaging data available from serial relapses; demonstrating that the majority of end-stage lesions were not detectable until the last progression event before death and

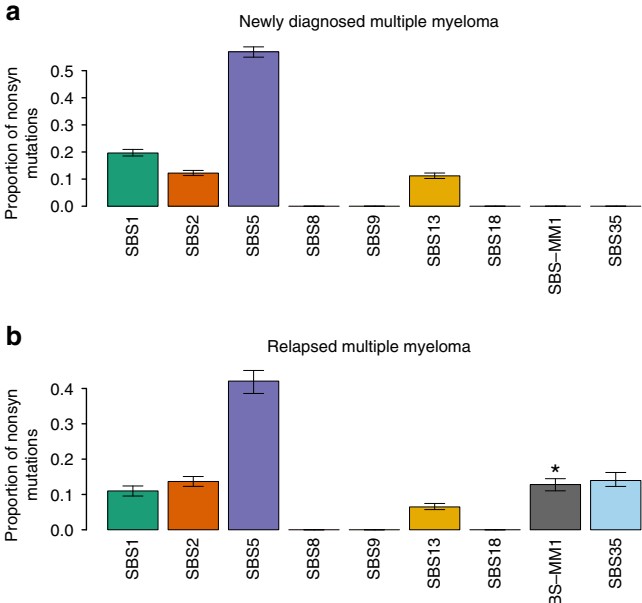

**Fig. 5 The mutational signature landscape of nonsynonymous mutations in MM.** Contribution of each MM mutational signature among nonsynonymous (nonsyn) mutations at diagnosis (**a**) and relapse (**b**). (SBS-MM1 = melphalan-associated signature; SBS35 = platinum-based chemotherapy associated signature). The asterisk reflects the presence of transcriptional strand bias for SBS-MM1. Confidence interval of each mutational signature was generated by drawing 1000 mutational profiles from the multinomial distribution, each time repeating the signature fitting procedure, and finally taking the 2.5th and 97.5th percentile for each signature.

subsequent postmortem examination (Fig. 7; Supplementary Data 2).

In I-H-130718, the time lag between the first and the second round of high-dose melphalan therapy was 25 months (Fig. 2b). In this short time window, we show that a single MM propagating cell survived exposure to high-dose melphalan and its subsequent dissemination drove relapse (Figs. 5 and 6). Then, the exposure of cells at these diverse sites when treated with both a platinum-containing regimen and high-dose melphalan, increased their mutational burden (Figs. 4a,e and 6c). The systemic seeding in I-H-130720 did not fit in the above described single-cell expansion model, having detectable SBS-MM1 only in two out of five branches (Fig. 6a). This distribution, together with the short survival and relapsed/refractory disease might reflect either the absence of a single cell expansion post-melphalan in some pre-existing disease localizations or the engraftment of clones re-infused with the autologous stem cell transplant[21]. The presence of unique chemotherapy and non- chemotherapy related mutations in all I-H-130720 branches is in contrast with the absence of single cell expansion (Figs. 3 and 6). The similar mutational burden of chemotherapy-related signatures across different branches in both I-H-130718 and I-H-130720 is in line with the synchronous exposure of these surviving cells to the same genotoxic agent (Fig. 6a).

Overall, these data suggest that a single cell has the potential to drive disease progression, and that systemic seeding of a single clonal cell can occur rapidly at relapse. To further investigate this hypothesis and to explore potential differences between pre- and post-treatment disease seeding, we quantified

the differential contribution of mutational signatures associated with aging (SBS1 and SBS5, described as clock-like) between the branches and the trunk[17,30,31]. The SBS5 profile has significant overlap with both SBS-MM1 and SBS35, and this can lead to the incorrect assignment of signatures, one of the major issues in mutational signature analysis[18]. To avoid this, we focused on the ratio of SBS1 between branches and the trunk in each patient included in the WGS and the WXS cohorts. The ratio for each patient was corrected for the number of evolutionary trajectories, avoiding the pooling of mutations that were acquired in parallel. Interestingly, the SBS1 ratios were significantly higher in the treatment-naive patients compared to those observed in the relapsed WXS and WGS cases (Fig. 8a and Supplementary Fig. 6), consistent with a subclonal diversification having occurred over a long period of time. These findings support the model in which MM seeding can be accelerated following high-dose treatment in comparison to that seen during spontaneous evolution. This acceleration and rapid development of myeloma lesions at relapse is supported by the PET/CT imaging data collected over time, where the majority of the investigated lesions occurred within a short time window (Fig. 7).

## Discussion

In this study, we used multiple concurrent samples obtained from several rarely biopsied anatomical sites, obtained by warm autopsy in patients with relapsed/refractory MM. This unique sample source and study design allowed us to interrogate the spatio-temporal genomic heterogeneity of MM at the time of aggressive relapse in a set of patients previously exposed to high-dose melphalan. Investigating the mutational signature landscape, we showed the strong mutagenic activity of melphalan and platinum-based agents on the MM propagating cell and its contribution to the post relapse genomic landscape. While these mutations tend to occur in the late-replicating and non-coding parts of the genome[21], we provide evidence that exposure to chemotherapy is responsible for a considerable proportion (>20%) of the nonsynonymous mutations acquired at clinical relapse. Future larger studies in the post-autologous stem cell transplantation setting will examine the impact of these mutations on evolutionary trajectories, disease aggressiveness, late toxicities, and subsequent survival.

We demonstrate that MM seeding is promoted by an evolutionary process in which distinct clones harboring distinct drivers are selected and expanded at varying anatomic sites. Using chemotherapy-related mutational signatures as a genomic barcode, we demonstrate that this complex process can be driven by a single surviving cell, potentially able to disseminate throughout the entire body. Linking these mutational signatures with the documented timing of chemotherapy exposure, we showed that, at clinical relapse, systemic seeding of MM can occur in a very short time window. Importantly, the patterns we demonstrate at relapse are strikingly different to the spatio-temporal patterns of evolution and selection that have been demonstrated during spontaneous evolution prior to the time of initial diagnosis and exposure to therapy. In fact, at diagnosis, different anatomic disease sites are characterized by high burden of clock-like mutation (i.e., SBS1)[17,21,30], consistent with an early divergence from the most common recent ancestor followed by slow growth (Fig. 8a). The accelerated anatomic dissemination we describe at relapse is similar to the metastatic seeding recently reported in different solid cancers[31,32]. This process is likely the result of a combination of two factors: (1) the selection of a more aggressive/proliferative

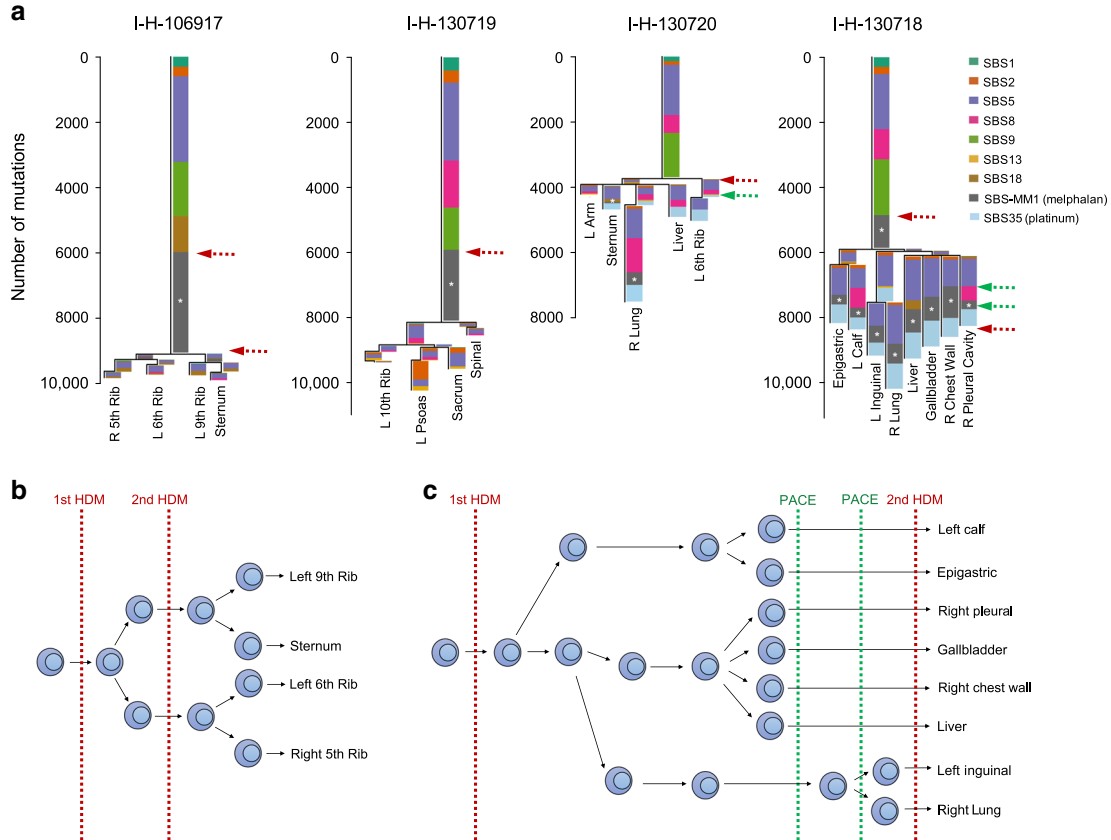

**Fig. 6 Timeline of mutational signatures. a** Mutational signature contribution for each phylogenetic tree cluster for each sample. Asterisks indicate the presence of transcriptional strand bias for SBS-MM1. Red and green dashed arrows represent exposure to melphalan and platinum-based therapies, respectively. L = left; R = right. B-cc) A schema summarizing the relationship between chemotherapy, subclonal selection and seeding in (**b**) I-H-106917 and (**c**) I-H-130718. (HDM = high-dose melphalan, PACE = cisplatin, doxorubicin, cyclophosphamide and etoposide, SBS = single base substitution). Red and green dashed lines in (**b**) and (**c**) represent exposure to melphalan and platinum-based therapies, respectively.

clones, and (2) treatment-related immunosuppression. While the first factor is often unpredictable and undetectable with the current bulk sequencing technologies, the second factor represents a rational approach to improve treatment efficacy. Indeed, strategies encouraging immune reconstitution may prevent this acceleration and reduce the incidence of clinical relapse.

These data highlight the importance of considering the complex spatial and temporal heterogeneity of MM in the evaluation of treatment-response and in minimal residual disease assessment and provide a strong rationale for comprehensive characterization of the MM genomic complexity to enhance clinical decision making.

## Methods

**Patient characteristics**. Twenty-one tumor and four non-tumor samples were collected from four patients enrolled in the Last Wish Program at MSKCC (Fig. 2a; Supplementary Table 1)[25]. The Last Wish Program is an ongoing research biospecimen protocol (protocol #15-021, approved by the Institutional Review Board of MSKCC) which permits the postmortem collection of tissue and other samples from deceased patients, from whom consent for the protocol was obtained antemortem. The protocol allows for the performance of a broad range of research studies using the collected samples. All patients were treated with multiple lines of therapy (median 6, ranges 5–8) including combinations of novel agents. All patients had previously received at least one round of high-dose melphalan therapy followed by autologous stem cell transplant (Fig. 2b). Two patients had an overall survival longer than 7 years (I-H-106917 and I-H-130719); in contrast, the other two died within 3 years of diagnosis (I-H-130718 and I-H-130720) (Fig. 2b).

All available 18F-fluorodeoxyglucose (FDG) PET/CT imaging studies for each patient were reviewed by a dual Diagnostic Radiology and Nuclear Medicine board certified radiologist with 15 years of FDG PET/CT experience (G.A.U.). Maximum intensity projection images of FDG-PET/CT studies and maximum standardized uptake values (SUVmax) of reference lesions were obtained using PET VCAR (GE Healthcare).

**Sequencing and genomic analysis**. Each tumor sample was collected from a different disease localization site (Fig. 2a) and DNA was extracted from CD138+ purified cells. To avoid contamination related to late systemic disease dissemination, cells collected from skeletal muscles were used as matched normal controls. All normal samples were histologically reviewed to exclude microscopic foci of tumor. Tumor biopsies collected were commonly very cellular and only those with >70% cellularity based on histologic review were selected for DNA extraction. After PicoGreen quantification and quality control by Agilent BioAnalyzer, 500 ng of genomic DNA were sheared using a LE220-plus Focused-ultrasonicator (Covaris catalog # 500569) and sequencing libraries were prepared using the KAPA Hyper Prep Kit (Kapa Biosystems KK8504) with modifications. In brief, libraries were subjected to a 0.5X size select using aMPure XP beads (Beckman Coulter catalog # A63882) after post-ligation cleanup. Libraries not amplified by PCR (07652_C) were pooled equivolume and were quantitated based on their initial sequencing performance. Libraries amplified with 5 cycles of PCR (07652_D, 07652_F, 07652_G) were pooled equimolar. Samples were run on a NovaSeq 6000 in a 150 bp/150 bp paired end run, using the NovaSeq 6000 SBS v1 Kit and an S4 flow cell (Illumina).

The median coverage for tumor and normal samples was 92.1X and 58.8X respectively (Supplementary Table 2). All bioinformatics analyses were performed using our in-house pipeline Isabl (Medina et al., in preparation). In brief, FASTQ files were aligned to the GRCh37 reference genome using BWA-MEM, and de-duplicated aligned BAM files were analyzed using the following published tools: (1) Battenberg for clonal and subclonal CNAs[33]; (2) BRASS for SVs[34]; (3) CaVEMan and Pindel for SNVs and small indels[35,36]. The clonal

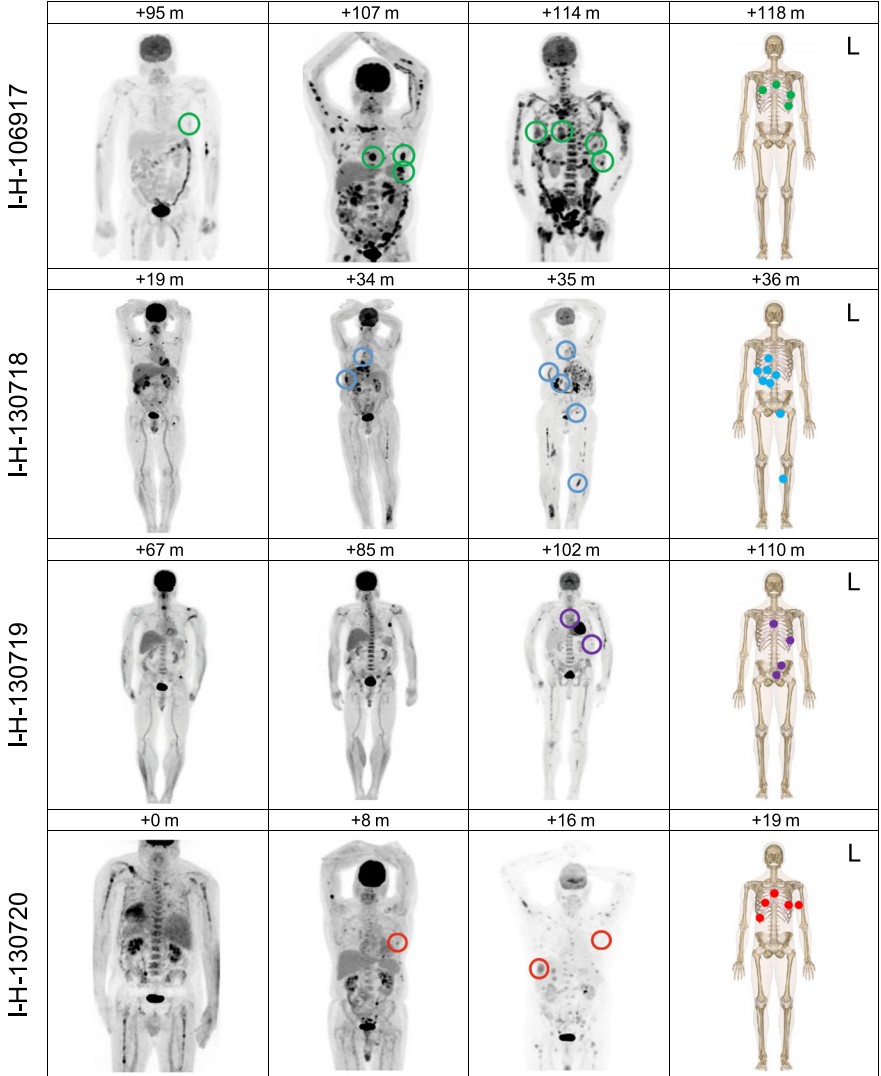

**Fig. 7 Tracking of biopsied lesions through the disease course by FDG-PET/CT.** Biopsy sites and FDG-PET/CT correlates on Maximum Intensity Projection images for each patient arranged by the number of months following diagnosis (m = months). All included scans show phases of active disease. Biopsy sites were only annotated on FDG-PET/CT if identified on Radiologist review. L = left.

composition and phylogenic tree of each MM patient was reconstructed by running the DP[8,33]. Only clusters with more than 50 mutations were considered (as recently described)[8,14]. In the only patient that underwent allogeneic stem cell transplant (I-H-130718), all samples had a subclonal DP cluster of 1594 mutations with a median cancer cell fraction <10%, reflecting the unique donor single-nucleotide polymorphism (SNP) profile. These mutations were not included in any subsequent analysis.

Three main classes of complex SVs were observed in MM: chromothripsis, templated insertion and chromoplexy and defined according to the most recent criteria[14,27,37,38].

**Mutational signatures.** Mutational signature analysis was performed applying our recently published workflow, based on three main steps: de novo extraction, assignment and fitting[18]. For the first step, we ran *SigProfiler*[17] combining our multi-spatial WGS cohort with a recently published cohort of 52 WGS patients[9,14,21,39]. Then all extracted signatures were assigned to the latest COSMIC reference (https://cancer.sanger.ac.uk/cosmic/signatures/SBS/) in order to define which known mutational processes were active in our cohort. Finally, we applied our recently developed fitting algorithm (*mmsig*) to confirm the presence and estimate the contribution of each mutational signature in each sample[21]. Confidence intervals were generated by drawing 1000 mutational profiles from the multinomial distribution, each time repeating the signature fitting procedure, and finally taking the 2.5th and 97.5th percentile for each signature. Mutational signature transcriptional strand bias analysis was

performed using *SigProfiler* and integrated into *mmsig*. The source code of *mmsig* is available on GitHub: https://github.com/evenrus/mmsig.

**Validation set.** We imported recently published WXS data obtained from multiple samples (N = 125) collected from different anatomic loci from both newly diagnosed and relapsed MM patients (EGAS00001002111, n = 51)[13]. In 40 patients, multiple samples were collected at diagnosis from different sites; in the other 11 patients, at least one sample was collected at relapse after intensive treatment, such as platinum-containing regimens and high-dose melphalan with autologous stem cell transplant (Supplementary Tables 3–4). In total, 125 tumor and 51 normal WXS data were included in this study. FASTQ files were aligned to the reference genome using BWA-MEM. BAM files were analyzed for SNV and indels using CaVEMan and Pindel, similarly to the WGS cohort[35,36]. The CNA profile of each sample was estimated using Facets[40]. The clonal and subclonal architecture of each patient was reconstructed using the DP for 47 patients[7]. In four patients, the DP failed due to either low CNA quality or the low sample purity. These patients were removed from the study.

**Data analysis and statistics.** Data analysis was carried out in R version 3.6.1. Standard statistical tests are mentioned consecutively in the manuscript while more complex analyses are described above. All reported *p*-values are two-sided, with a significance threshold of <0.05.

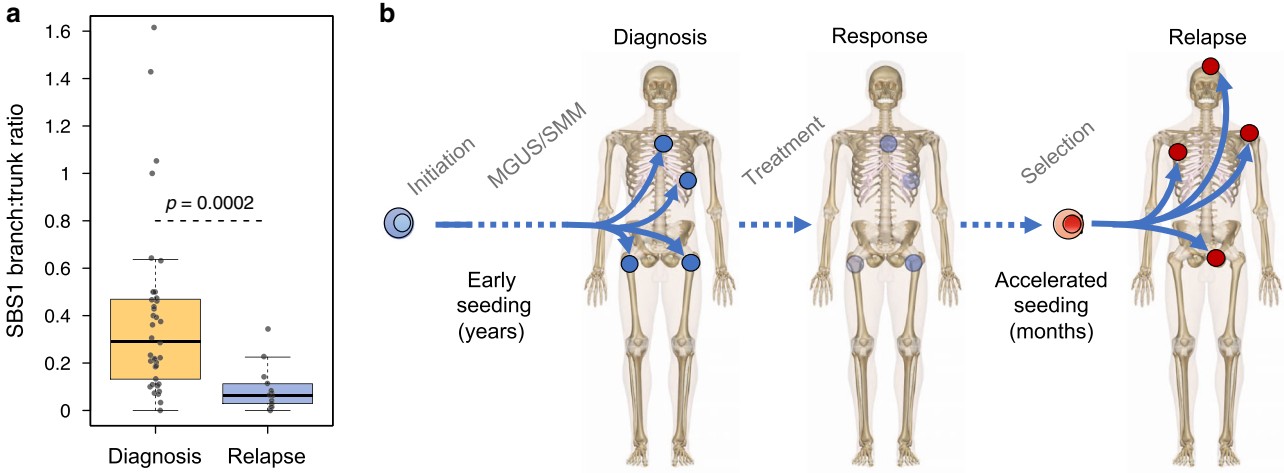

**Fig. 8 Early versus late divergence of disease sites. a** SBS1 is a mutational signature reflective of biological cell aging. By estimating changes in SBS1 in the trunk and the branches, we were able to study the speed of the tumor evolution in a given site at a given time-point. Here, we show the difference in SBS1 branch:trunk ratio between newly diagnosed and relapsed multiple myeloma (*p*-value estimated using Wilcoxon tests). A lower SBS1 branch:trunk ratio represents a highly accelerated evolution since divergence from the trunk (the most recent common ancestor). Boxplots show the median and interquartile range; observations outside this interval are shown as dots. **b** A schema summarizing the seeding patterns over time, starting as slow seeding and tumor growth during the precursor phase, with acceleration in advanced disease.

**Reporting summary**. Further information on research design is available in the Nature Research Reporting Summary linked to this article.

## Data availability

Sequence data that support the findings of this study have been deposited in European Genome-phenome archive under the Accession codes EGAS00001002111 and EGAS00001004404

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

## Acknowledgements
This work is supported by the Memorial Sloan Kettering Cancer Center NCI Core Grant (P30 CA 008748) and by Susan and Peter Solomon Divisional Genomics Program. FM is supported by the American Society of Hematology, the International Myeloma Foundation and The Society of Memorial Sloan Kettering Cancer Center. K.H.M. is supported by the Haematology Society of Australia and New Zealand New Investigator Scholarship and the Royal College of Pathologists of Australasia Mike and Carole Ralston Travelling Fellowship Award. G.J.M. is supported by The Leukemia Lymphoma Society. C.I.D. is supported by NCI grants R35 CA220508 and U2C CA233284 and the Kleberg Foundation. J.U.P. reports funding from NHLBI NIH Award K08HL143189 and the Parker Institute for Cancer Immunotherapy at Memorial Sloan Kettering Cancer Center.

## Author contributions
H.L., F.M., E.P., C.I.D., and O.L. designed and supervised the study, collected and analyzed data and wrote the paper. V.Y. collected and analyzed data and wrote the paper. B.T.D., E.H.R., K.H.M., G.G., J.M.M., J.A.O., M.L., and Y.Z. analyzed data. R.K., P. Blaney., M.A., A.M.M., L.Z., M.P., E.B., C.A., Y.Z.H.A., A.D., D.C., S.G., O.B.L., J.U.P., M.S., G.S., H.H., M.L.H., A.M.L., S.L., N.S.K., S.M., E.S., U.S., P. Baez, F.V.R., G.A.U., and G.J.M. collected the data.

## Competing interests
J.U.P. reports research funding, intellectual property fees, and travel reimbursement from Seres Therapeutics and consulting fees from DaVolterra. M.S. has served as a paid consultant for McKinsey & Company, Angiocrine Bioscience, Inc., and Omeros Corporation. He has received research funding from Angiocrine Bioscience, Inc. He has served on an ad hoc advisory board for Kite – A Gilead Company. H.J.L. has received research support for clinical trials from Takeda. She served on the advisory boards of Takeda, Celgene, Janssen, Sanofi, Caelum Biosciences and has been a consultant for Karyopharm and Pfizer. Other authors declare no competing interests.
