## [Peer Review File · Nature Communications]

REVIEWER COMMENTS

Reviewer #1 (Remarks to the Author): Expert in MM

Multiple myeloma (MM) will typically present as an indolent malignancy characterized by patchy bone marrow infiltration and slow replication. Continued mutagenesis results in clonal heterogeneity and frequently, aggressive relapse with exponential growth involving both extra and intramedullary sites. Landau et al. present data to suggest that aggressive relapse may follow emergence of a single dominant subclone capable of widespread dissemination. This is a novel, if unsurprising finding.

The authors report whole genome sequencing (WGS) of 4 patients who succumbed to relapsed MM. Specifically, WGS was performed on DNA isolated from CD138-selected myeloma cells taken at warm autopsy from disparate anatomic sites - average 5 soft tissue or bony plasmacytomas per cadaver. Skeletal muscle DNA was used as control. Somatic mutations were identified in all plasmacytoma samples (median 10938 mutations) and cataloged according to mutational signature using established algorithms. Comparison of sequences from the same cadaver allowed investigators to label mutations as clonal or subclonal, and to trace the genomic evolution of each plasmacytoma. All 4 patients had received high dose melphalan, and unique mutations believed attributable to that exposure were identified in samples taken from all 4 cadavers. In 3 of the 4, every plasmacytoma sampled bore an identical set of melphalan-associated mutations unique to that patient. It is this finding which suggests that relapse can be fueled by emergence of a single dominant subclone, melphalan exposure providing a convenient timestamp.

The concept is elegant. The results provide insight into myeloma progression, and will be of considerable interest to that field, although prevalence among MM patients is difficult to predict given the small sample size. Of less interest to a broader audience as these findings are in line with those seen for other malignancies, and offer little mechanistic insight. Longitudinal sample data is lacking, although the authors do provide whole exome sequencing data derived from 51 MM patients at diagnosis or relapse.

The manuscript is succinct, and for the most part well written, however there are aspects that are unnecessarily confusing. For example, the abbreviation SBS-MM1 is used to describe both the unique set of somatic mutations identified in a given clone that have been attributed to melphalan exposure, as well as the molecular signature that enables such classification. Inclusion of supplemental figures 3 and 4 in the main text would help clarify. Should additional space be required, figure 1 could be incorporated into figure 7, and figure 6 eliminated.

editing:

Abbreviations should be defined when first introduced

Figure 5 I-H-130720 – green arrow to denote platinum exposure is missing

Figure legends to supplemental figures 3 and 4 are incorrect: left and right are reversed

Reviewer #2 (Remarks to the Author): Expert in MM

In their manuscript, Landau and colleagues attempt to systematically investigate the genomic evolution of multiple myeloma (MM) at spatially distinct sites over time. In order to clarify whether relapse of MM at different anatomic sites is due to secondary seeding from other disease locations or expansion of pre-existing, previously undetectable clones, the authors leveraged the unique signature of chemotherapy-related mutagenic processes to link clonal expansion of MM cells to a discrete time window in each patient's disease history. The authors provide evidence of accelerated MM seeding at clinical relapse is primarily driven by clonal expansion of MM cells that survived treatment and acquired additional mutations associated with high dose melphalan and

cisplatin utilizing samples from various metastatic sites in 4 recently deceased patients that underwent rapid autopsies.

Major comments:

1. The major focus of the manuscript appears to be the significant impact of cytotoxic chemotherapy such as melphalan and cisplatin in generating additional mutations. However, melphalan has clearly been shown to favorably impact the overall survival of patients. In fact, despite the potential for generating novel mutations, the time frame from treatment to the appearance of these mutations in these 4 patients ranges from a few months to more than 8 years. In these 2 long-term survivors, what is the biologic evidence that there is a direct causal relationship between melphalan and mutational burden considering the time lag?
2. As written, it is unclear what the true biologic significance is of acquired mutations. Does this lead to the selective clonal expansion of certain clones? Is there an equal or unequal distribution of these branch clones in the different metastatic sites? Do the development of the metastatic sites appear to be synchronous or dyssynchronous?
3. Supplemental figures 3 and 4 nicely highlight fundamental differences in genome plots between long-term and short-term survivors. Would recommend moving these figures into the main text of the manuscript.
4. In line 200, a more thorough explanation for this phenomenon in pt. I-H-106917 should be provided in order to clarify the loss of SBS-MM1 detection in 2nd/3rd level branches. For instance, if no dominant subclone was to develop, newly acquired chemotherapy-related signatures would not reach the threshold for detection with WGS. This could be associated with achievement of less than a complete response and the lack of a genetic "bottleneck" phenomenon. Although a larger dataset may be needed to provide a definitive answer, authors could better discuss these findings and provide some correlation with the clinical outcome/treatment history of this patient.
5. Systemic seeding in pt. I-H-130720 should be better explained and correlated with WGS and clinical data (lines 227-232). The absence of single cell expansion s/p MEL therapy in some localizations could reflect the persistence of multiple subclones with an overall preserved clonal structure.
6. In their evaluation of spatio-temporal genomic heterogeneity of MM, the authors exploit SBS signatures associated with melphalan and platinum exposure to define evolutionary trajectories. Although it may not be feasible in this cohort of patients, a brief discussion on the possible role of immunomodulatory drugs and response depth in MM clonal and subclonal evolution should be provided, especially in light of the findings reported by Jones et al., *Haematologica* 2019, <https://doi.org/10.3324/haematol.2018.202200>.

Minor comments:

1. In the Introduction, when referring to SBS, it may be relevant to cite Alexandrov et al., *Nature* 2020, <https://doi.org/10.1038/s41586-020-1943-3>.
2. In the Results section, line 126, the anatomic location of non-tumor samples should be specified in the main text. In line 130, the reference to the source of the WXS cohort data should be explicit. In line 134, "Genomic analysis and Validation set paragraphs, Methods section" should be added instead of just "(Methods)".
3. In line 145, a more thorough explanation on Fig. 3 should be provided and Figure 3 should be labeled adequately in order to clarify which patient has either long-/short- survival.

Reviewer #3 (Remarks to the Author): Expert in genomics

In this study, Landau et al have investigated the topographic and temporal clonal evolution of Multiple Myeloma (MM) by sequencing the whole genome of 21 tumor samples from different sites and matched normal samples obtained at the autopsy of four patients who have received different treatments including melphalan and platinum. The authors also reanalyze the whole exome sequences of 125 samples (51 patients) previously published as a validation cohort. The authors used two mutational signatures associated with treatment with melphalan and platinum to reconstruct the subclonal evolution of the tumor. The study identifies that in 3 of the 4 patients analyzed by WGS all multiple tumor sites seem to derive from a single cell resistant to melphalan therapy that emerges after this therapy. This clone further diverges at the different sites accumulating private mutations. In the remaining case, the multiple sites seem to derive from a clone that emerges previously to melphalan therapy, probably present in the auto transplant sample. The authors also determine that the seeding at relapse occurs rapidly compared to the previous evolution of the tumor.

The study provides novel insights to the dynamic evolution of MM that may have implications for designing therapeutic strategies. The authors should address some minor aspects that may help to better clarify the study

1. Supplementary tables with the alterations (SNV, CNA and SV) identified in each sample from both cohorts (WGS and WXS) should be provided
2. Introduction, line 110. It would be clearer if you specify that you refer to stem cells in auto-transplant. Is not this the idea?
3. Introduction, line 112: "SBS-MM1 represents unique single-cell genomic barcode for clonal cells derived from a single propagating cell". I would not limit this idea to SBS-MM1 signature considering that other chemotherapy-related signatures might be used also as "genomic barcodes", as also described within the manuscript for the platinum-related signature.
4. Figure 2B provides the timeline of treatments of the four patients. However, the moment in which the auto or allo-transplants are performed is not provided. Given that this intervention seems relevant to interpret the patterns observed in patient IH-130720 it would be important to show also these treatments in the figure.
5. Figure 5A. I am afraid I do not understand why the authors include the arrow indicating melphalan in case I-H-130720 at a point that there is not a melphalan signature in the tree. It is not clear.
6. Figure 5. Is it possible that the representation in panel C does not fit exactly with the phylogenetic tree shown in panel A (I-H-130718)? Why the liver biopsy/clone is separated from the Right pleural, Gallbladder, and Right chest wall clones so early? This is not what is shown in the phylogenetic tree that all the clones seem to branch at the same time. Besides, the presence of SBS35 in the branch before splitting Left inguinal and Right lung in panel A suggests that first PACE line of treatment should took place before the separation of these two subclones in panel C.
7. Supplementary Figure 3. The order of "trunk" and "branches" is not consistent in figure and figure legend. This reviewer thinks that the correct name for these plots is "circos" and not "circus" as written in the figure legend (also in Supplementary Figure 4).
8. Supplementary Figure 5 (Excel file): Is the label of this table correct? Besides, a "yes" in First PET-TC -> Results CT should be "pos".
9. Writing might be improved in some sentences. For instance:

- Abstract, line 56: "To address this, we interrogated 25 samples, by whole genome sequencing, collected at autopsy...".
- Results, line 126-127, "we investigated the WGS profile of twenty-one tumor and four non-tumor samples were collected from four patients.". This "were" should be removed.
- Gene names should be written in italics in Figure 3.
- Results, line 191-192. The second part of the sentence is not very clear.
- Discussion, line 267-268: "we demonstrate that how this complex process can be driven...". Please, check this "that how".

REVIEWER COMMENTS

Reviewer #1 (Remarks to the Author): Expert in MM

Multiple myeloma (MM) will typically present as an indolent malignancy characterized by patchy bone marrow infiltration and slow replication. Continued mutagenesis results in clonal heterogeneity and frequently, aggressive relapse with exponential growth involving both extra and intramedullary sites. Landau et al. present data to suggest that aggressive relapse may follow emergence of a single dominant subclone capable of widespread dissemination. This is a novel, if unsurprising finding.

The authors report whole genome sequencing (WGS) of 4 patients who succumbed to relapsed MM. Specifically, WGS was performed on DNA isolated from CD138-selected myeloma cells taken at warm autopsy from disparate anatomic sites - average 5 soft tissue or bony plasmacytomas per cadaver. Skeletal muscle DNA was used as control. Somatic mutations were identified in all plasmacytoma samples (median 10938 mutations) and cataloged according to mutational signature using established algorithms. Comparison of sequences from the same cadaver allowed investigators to label mutations as clonal or subclonal, and to trace the genomic evolution of each plasmacytoma. All 4 patients had received high dose melphalan, and unique mutations believed attributable to that exposure were identified in samples taken from all 4 cadavers. In 3 of the 4, every plasmacytoma sampled bore an identical set of melphalan-associated mutations unique to that patient. It is this finding which suggests that relapse can be fueled by emergence of a single dominant subclone, melphalan exposure providing a convenient timestamp.

The concept is elegant. The results provide insight into myeloma progression, and will be of considerable interest to that field, although prevalence among MM patients is difficult to predict given the small sample size. Of less interest to a broader audience as these findings are in line with those seen for other malignancies, and offer little mechanistic insight. Longitudinal sample data is lacking, although the authors do provide whole exome sequencing data derived from 51 MM patients at diagnosis or relapse.

The manuscript is succinct, and for the most part well written, however there are aspects that are unnecessarily confusing. For example, the abbreviation SBS-MM1 is used to describe both the unique set of somatic mutations identified in a given clone that have been attributed to melphalan exposure, as well as the molecular signature that enables such classification. Inclusion of supplemental figures 3 and 4 in the main text would help clarify. Should additional space be required, figure 1 could be incorporated into figure 7, and figure 6 eliminated.

We thank the reviewer for his/her comments and comprehensive summary of our work. Following his/her comments:

- 1) We have adjusted the terminology throughout the paper; using SBS-MM1 only to denote the single-base substitution signature associated with melphalan-exposure.
- 2) As suggested by both Reviewer #1 and #2, we have moved **Supplementary Figure 3** and **4** into the main text (now combined in a new **Figure 4**)

editing:

Abbreviations should be defined when first introduced

We carefully revised the text and defined all abbreviations.

Figure 5 I-H-130720 – green arrow to denote platinum exposure is missing

The missing arrow has now been included.

Figure legends to supplemental figures 3 and 4 are incorrect: left and right are reversed

Supplementary Figure 3 and 4 have been merged and included in the updated version as **Figure 4**. The legend has been corrected.

Reviewer #2 (Remarks to the Author): Expert in MM

In their manuscript, Landau and colleagues attempt to systematically investigate the genomic evolution of multiple myeloma (MM) at spatially distinct sites over time. In order to clarify whether relapse of MM at different anatomic sites is due to secondary seeding from other disease locations or expansion of pre-existing, previously undetectable clones, the authors leveraged the unique signature of chemotherapy-related mutagenic processes to link clonal expansion of MM cells to a discrete time window in each patient's disease history. The authors provide evidence of accelerated MM seeding at clinical relapse is primarily driven by clonal expansion of MM cells that survived treatment and acquired additional mutations associated with high dose melphalan and cisplatin utilizing samples from various metastatic sites in 4 recently deceased patients that underwent rapid autopsies.

Major comments:

1. The major focus of the manuscript appears to be the significant impact of cytotoxic chemotherapy such as melphalan and cisplatin in generating additional mutations. However, melphalan has clearly been shown to favorably impact the overall survival of patients. In fact, despite the potential for generating novel mutations, the time frame from treatment to the appearance of these mutations in these 4 patients ranges from a few months to more than 8 years. In these 2 long-term survivors, what is the biologic evidence that there is a direct causal relationship between melphalan and mutational burden considering the time lag?

In our recent paper "Timing the initiation in multiple myeloma" by Rustad et al (doi: <https://doi.org/10.1038/s41467-020-15740-9>), we demonstrated a causative correlation between melphalan and the SBS-MM1 signature, with SBS-MM1 only being detected in patients who had relapsed following melphalan exposure. If SBS-MM1 is directly caused by melphalan, the majority of SBS-MM1-associated mutations (detected either in the trunk or in the branches) were acquired as a consequence of melphalan exposure. This causative association provides a link between distinct phylogenetic tree

units and distinct time windows in each patient's life (i.e. the time when chemotherapy was administered). The same concept can be used for SBS35 (a platinum-associated signature) as recently done by Pinch et al Nat Gen 2019. Considering this evidence and data, we can conclude that these mutations appeared (or were acquired) at the time of transplant. We are not stating that the chemotherapy-associated mutations developed more than 8 years following exposure, rather we demonstrate that at the time of the autopsy these mutations were still in the cancer cells, and were therefore selected and detected by WGS.

The different time-lag is likely dependent on other genomic and clinical features. SBS-MM1-associated mutations did not emerge as key drivers in cancer progression and in defining clinical outcome. While the role of these chemotherapy-related mutations is unclear, their presence can be used to link and reconstruct the temporal and spatial evolution of each patient, and this is the main aim of our study.

We acknowledge that high-dose melphalan and autologous transplant represents one of the gold standard treatments in young and fit multiple myeloma patients, and our paper doesn't have the statistical power to suggest any definitive warning about its usage. It is logical that for many patients any negative impact of treatment-induced mutations will be out-weighed by the clinical benefit of transplant, however in some patients it is possible that the induced mutations could be of increased clinical relevance. Larger cohorts are needed to evaluate the clinical impact of these mutations on disease aggressiveness and subsequent survival in the post-transplant setting. This important point has now been included in the new version of the manuscript (**Pages 12-13; Lines 273-275**).

2. As written, it is unclear what the true biologic significance is of acquired mutations. Does this lead to the selective clonal expansion of certain clones? Is there an equal or unequal distribution of these branch clones in the different metastatic sites? Do the development of the metastatic sites appear to be synchronous or dyssynchronous?

We know from previous studies (e.g. Bolli et al Nat Comm 2014 and 2018; Jones et al Haematologica 2019) that myeloma progression after treatment is led by complex Darwinian evolutionary models where genomic drivers are selected or acquired conferring proliferation advantage. The post-melphalan increased number of nonsynonymous mutations might play a role in leading the progression, however the power of our study doesn't allow such speculation. As recently described (Maura et al Nat Comm 2019) the clonal expansion and seeding seem to be driven by selection of structural variants, copy number changes and driver mutations. The first two seem to have a major role, both in the cited paper and this current manuscript. In the updated version of the manuscript this concept has now been included and better explained (Page 7; line 156-160). Furthermore, the entire catalogue of key drivers in each sample is now included as **Supplementary Figure 5**.

All but I-H-130720 showed similar chemotherapy-related and unrelated mutational burden across the phylogenetic tree. This is in line with the synchronous exposure of tumor cell to the same agent (e.g. melphalan). This part has now been included in the updated manuscript (**Page 11; Lines 238-241**).

Due to the bleeding of SBS5 and SBS-MM1/SBS35 it is hard to apply in these cases our recent molecular clock workflow and estimate the time of clonal expansion at each site (Rustad H et al Nat Comm 2020). However, examining radiological images over time (PET-CT; new **Figure 7**), it seems that most of the lesions expanded in a similar time window. However, the exact moment in which these lesions first occur is impossible to say. This data has now better described in the updated version of the manuscript (**Page 12; Lines 258-260**).

3. Supplemental figures 3 and 4 nicely highlight fundamental differences in genome plots between long-term and short-term survivors. Would recommend moving these figures into the main text of the manuscript.

Following Reviewers suggestions, in the updated version of the manuscript, we combined **Supplemental figures 3 and 4** and moved them into the main text as **Figure 4**.

4. In line 200, a more thorough explanation for this phenomenon in pt. I-H-106917 should be provided in order to clarify the loss of SBS-MM1 detection in 2nd/3rd level branches. For instance, if no dominant subclone was to develop, newly acquired chemotherapy-related signatures would not reach the threshold for detection with WGS. This could be associated with achievement of less than a complete response and the lack of a genetic “bottleneck” phenomenon. Although a larger dataset may be needed to provide a definitive answer, authors could better discuss these findings and provide some correlation with the clinical outcome/treatment history of this patient.

We agree with the reviewer that absence of selection would hide SBS-MM1-associated mutations. However, all described branches are characterized by unique drivers and genomic alterations (e.g. SVs) reflecting the existence of previous selection. These selected clones share the same SBS-MM1 signature suggesting that they come from the same single cell survived and selected after high dose melphalan. This concept has been included and better explained in the updated version of the manuscript (**Page 11; Lines 235-238**).

It is true that clinical outcome and response quality have been reported to be associated with slightly different patterns of progression. These studies have been done comparing samples collected before and immediately after treatment. In our study we investigated only end-stage samples which reflected the sum of multiple selections that each tumor experienced. Without samples collected between different therapies and progressions it is impossible to provide any clear clinical and translational conclusion.

5. Systemic seeding in pt. I-H-130720 should be better explained and correlated with WGS and clinical data (lines 227-232). The absence of single cell expansion s/p MEL therapy in some localizations could reflect the persistence of multiple subclones with an overall preserved clonal structure.

This is definitely a possible explanation of why a certain tumor don't show chemotherapy signatures despite previous exposure. However, all I-H-130720 samples

and branches showed unique set of SBS35 mutations and unique genomic aberrations, suggesting that a selection and single cell expansion happen at the end of patients life. Basing on the integration of platinum signatures and genomic drivers we think that the non-expanded model doesn't fit in this case. This concept is now included in the updated version of the manuscript (**Page 11, Lines 235-238**).

6. In their evaluation of spatio-temporal genomic heterogeneity of MM, the authors exploit SBS signatures associated with melphalan and platinum exposure to define evolutionary trajectories. Although it may not be feasible in this cohort of patients, a brief discussion on the possible role of immunomodulatory drugs and response depth in MM clonal and subclonal evolution should be provided, especially in light of the findings reported by Jones et al., *Haematologica* 2019, <https://doi.org/10.3324/haematol.2018.202200>.

Unfortunately, the lack of samples collected before and after each treatment doesn't allow to define any conclusion on this relevant point raised by the reviewer. (see point 4)

Minor comments:

1. In the Introduction, when referring to SBS, it may be relevant to cite Alexandrov et al., *Nature* 2020, <https://doi.org/10.1038/s41586-020-1943-3>.

We thank the reviewer for this suggestion- we had cited earlier publications. The most recent Alexandrov paper is now included.

2. In the Results section, line 126, the anatomic location of non-tumor samples should be specified in the main text.

The source of normal match samples has now been included at the beginning of the Results section (**Page 6; Line 128-129**)

In line 130, the reference to the source of the WXS cohort data should be explicit. In line 134, "Genomic analysis and Validation set paragraphs, Methods section" should be added instead of just "(Methods)".

The reference and link to the method section has now been included at the beginning of the Results section (**Page 6; Lines 135-136**)

3. In line 145, a more thorough explanation on Fig. 3 should be provided and Figure 3 should be labeled adequately in order to clarify which patient has either long-/short-survival.

The long-/short- survival information has been included in the figure legend.

Reviewer #3 (Remarks to the Author): Expert in genomics

In this study, Landau et al have investigated the topographic and temporal clonal evolution of Multiple Myeloma (MM) by sequencing the whole genome of 21 tumor samples from different sites and matched normal samples obtained at the autopsy of four patients who have received different treatments including melphalan and platinum. The authors also reanalyze the whole exome sequences of 125 samples (51 patients) previously published as a validation cohort. The authors used two mutational signatures associated with treatment with melphalan and platinum to reconstruct the subclonal evolution of the tumor. The study identifies that in 3 of the 4 patients analyzed by WGS all multiple tumor sites seem to derive from a single cell resistant to melphalan therapy that emerges after this therapy. This clone further diverges at the different sites accumulating private mutations. In the remaining case, the multiple sites seem to derive from a clone that emerges previously to melphalan therapy, probably present in the auto transplant sample. The authors also determine that the seeding at relapse occurs rapidly compared to the previous evolution of the tumor.

The study provides novel insights to the dynamic evolution of MM that may have implications for designing therapeutic strategies. The authors should address some minor aspects that may help to better clarify the study

1. Supplementary tables with the alterations (SNV, CNA and SV) identified in each sample from both cohorts (WGS and WXS) should be provided

In the new version of the manuscript we have summarized the myeloma driver landscape of our cohort in the new **Supplementary Table 5**.

2. Introduction, line 110. It would be clearer if you specify that you refer to stem cells in auto-transplant. Is not this the idea?

In the new version of the paper we specify that the third model could be the result of a myeloma cell engraftment with the stem cell reinfusion during autologous transplant.

3. Introduction, line 112: "SBS-MM1 represents unique single-cell genomic barcode for clonal cells derived from a single propagating cell". I would not limit this idea to SBS-MM1 signature considering that other chemotherapy-related signatures might be used also as "genomic barcodes", as also described within the manuscript for the platinum-related signature.

We agree with the reviewer that the single cell barcoding can be induced by any chemotherapy-related signatures. In the new version of the paper we have replaced "SBS-MM1" at this point with "chemotherapy-related mutational signatures".

4. Figure 2B provides the timeline of treatments of the four patients. However, the moment in which the auto or allo-transplants are performed is not provided. Given that this intervention seems relevant to interpret the patterns observed in patient IH-130720 it would be important to show also these treatments in the figure.

In the new version of the manuscript, the chemotherapy and radiotherapy exposure in each patient's life have been annotated with colored vertical arrow. The arrow colors are identical to **Figure 4** and an asterisk has been used to annotate the allogeneic stem cell transplant in IH-130718

5. Figure 5A. I am afraid I do not understand why the authors include the arrow indicating melphalan in case I-H-130720 at a point that there is not a melphalan signature in the tree. It is not clear.

Following reviewer's comments, we moved the melphalan-arrow below, linked to the branches where SBS-MM1 was detected.

6. Figure 5. Is it possible that the representation in panel C does not fit exactly with the phylogenetic tree shown in panel A (I-H-130718)? Why the liver biopsy/clone is separated from the Right pleural, Gallbladder, and Right chest wall clones so early? This is not what is shown in the phylogenetic tree that all the clones seem to branch at the same time. Besides, the presence of SBS35 in the branch before splitting Left inguinal and Right lung in panel A suggests that first PACE line of treatment should took place before the separation of these two subclones in panel C.

Following reviewer's comments, we revised and corrected the **Figure 5C** schematic phylogenetic tree.

7. Supplementary Figure 3. The order of "trunk" and "branches" is not consistent in figure and figure legend. This reviewer thinks that the correct name for these plots is "circos" and not "circus" as written in the figure legend (also in Supplementary Figure 4).

Following Reviewer #1 and #2 comments, we merged **Supplementary Figure 3 and 4** into the new **Figure 4**. The figure legend is now corrected.

8. Supplementary Figure 5 (Excel file): Is the label of this table correct? Besides, a "yes" in First PET-TC -> Results CT should be "pos".

The Supplementary Table labels and typos have been corrected (new **Supplementary Table 7**).

9. Writing might be improved in some sentences. For instance:

- Abstract, line 56: "To address this, we interrogated 25 samples, by whole genome sequencing, collected at autopsy..."
- Results, line 126-127, "we investigated the WGS profile of twenty-one tumor and four non-tumor samples were collected from four patients.". This "were" should be removed.
- Gene names should be written in italics in Figure 3.
- Results, line 191-192. The second part of the sentence is not very clear.
- Discussion, line 267-268: "we demonstrate that how this complex process can be

driven...". Please, check this "that how".

All sentences and figures have been changed/improved according to Reviewer suggestions.

REVIEWERS' COMMENTS:

Reviewer #1 (Remarks to the Author):

prior critiques have been adequately addressed.

editing: page 11 lines 239 through 244: discussion regarding the mutational pattern exhibited by IH130720, especially the double negative used in lines 240/241, seems unnecessarily confusing and might be reworded

Reviewer #2 (Remarks to the Author):

The authors have addressed the concerns raised with the initial review. No additional comments.

Reviewer #3 (Remarks to the Author):

The authors have addressed properly all comments and suggestions

REVIEWERS' COMMENTS:

Reviewer #1 (Remarks to the Author):

prior critiques have been adequately addressed.

editing: page 11 lines 239 through 244: discussion regarding the mutational pattern exhibited by IH130720, especially the double negative used in lines 240/241, seems unnecessarily confusing and might be reworded

The sentenced has been corrected.

Reviewer #2 (Remarks to the Author):

The authors have addressed the concerns raised with the initial review. No additional comments.

Reviewer #3 (Remarks to the Author):

The authors have addressed properly all comments and suggestions